# Characterization of the Astrocyte Calcium Response to Norepinephrine in the Ventral Tegmental Area

**DOI:** 10.3390/cells14010024

**Published:** 2024-12-30

**Authors:** Michele Speggiorin, Angela Chiavegato, Micaela Zonta, Marta Gómez-Gonzalo

**Affiliations:** 1Department of Biomedical Sciences, Università degli Studi di Padova, 35131 Padova, Italy; michele.speggiorin@phd.unipd.it (M.S.); angela.chiavegato@unipd.it (A.C.); 2Neuroscience Institute, Section of Padova, National Research Council (CNR), 35131 Padova, Italy; micaela.zonta@cnr.it; 3Padova Neuroscience Center, University of Padova, 35131 Padova, Italy

**Keywords:** ventral tegmental area, astrocyte, calcium, norepinephrine, IP3 receptor

## Abstract

Astrocytes from different brain regions respond with Ca^2+^ elevations to the catecholamine norepinephrine (NE). However, whether this noradrenergic-mediated signaling is present in astrocytes from the ventral tegmental area (VTA), a dopaminergic circuit receiving noradrenergic inputs, has not yet been investigated. To fill in this gap, we applied a pharmacological approach along with two-photon microscopy and an AAV strategy to express a genetically encoded calcium indicator in VTA astrocytes. We found that VTA astrocytes from both female and male young adult mice showed a strong Ca^2+^ response to NE at both soma and processes. Our results revealed that Gq-coupled α1 adrenergic receptors, which elicit the production of IP3, are the main mediators of the astrocyte response. In mice lacking the IP3 receptor type-2 (IP3R2^−/−^ mice), we found that the astrocyte response to NE, even if reduced, is still present. We also found that in IP3R2^−/−^ astrocytes, the residual Ca^2+^ elevations elicited by NE depend on the release of Ca^2+^ from the endoplasmic reticulum, through IP3Rs different from IP3R2. In conclusion, our results reveal VTA astrocytes as novel targets of the noradrenergic signaling, opening to new interpretations of the cellular and molecular mechanisms that mediate the NE effects in the VTA.

## 1. Introduction

Over the last years, the classical paradigm of functional brain circuits composed uniquely of networks of interacting neurons has been challenged by a growing body of evidence that supports the central role played by glial cell astrocytes in brain function. Nowadays, a network of highly interacting neurons and astrocytes is recognized to represent the fundamental module of information processing in the brain [1]. Astrocytes respond with cytosolic Ca^2+^ elevations to diverse neurotransmitters/neuromodulators such as glutamate, GABA, ATP, D-serine, acetylcholine, dopamine, serotonin, histamine, endocannabinoids, and norepinephrine [2]. Subsequently to Ca^2+^ mobilization, astrocytes release gliotransmitters, modify the activity of neurotransmitter transporters, and/or change the extracellular concentration of ions to modulate neuronal excitability and neuronal circuits [3,4,5,6]. Astrocyte Ca^2+^ transients elicited by neuronal mediators crucially depend on the activation of different G-protein-coupled receptors (GPCRs), known as metabotropic receptors, that are associated with diverse intracellular signaling pathways [2,7,8]. It is well known that the activation of Gq–GPCRs in astrocytes triggers the production of IP3 and the release of Ca^2+^ from the endoplasmic reticulum (ER), mainly through the activation of the IP3 receptor type-2 (IP3R2) Ca^2+^ channel [9]. Accordingly, a reduction of the Ca^2+^ response triggered by Gq-GPCRs is observed in IP3R2^−/−^ knockout mice [10]. Interestingly, in these mice also the Ca^2+^ transients triggered by Gi-GPCRs are reduced, suggesting that in astrocytes the activation of GPCRs classically coupled to Gi can activate IP3Rs through non-canonical mechanisms [1,11,12,13].

Norepinephrine (NE), also known as noradrenaline, is a monoamine neurotransmitter/neuromodulator that belongs to the group of catecholamines. In the brain, NE plays a key role in the regulation of vital functions such as arousal, sleep–wake cycle, attention, learning, memory, and stress [14,15,16]. Similarly to other monoamines, such as dopamine and serotonin, NE transmission does not always rely on one-to-one neuron synaptic transmission. Instead, the release of NE occurs frequently at varicosities without clear synaptic contacts [17], a form of intercellular communication called volume transmission. NE elicits its actions through the activation of three main GPCRs, α1, α2, and β adrenergic receptors (α1AR, α2AR, and βAR), associated with Gq, Gi, and Gs, respectively [7]. The diffusion of NE throughout the extracellular space during asynaptic volume transmission facilitates the access of this neuromodulator to non-neuronal cells such as astrocytes [18,19]. Accordingly, results from both in vitro and in vivo preparations revealed that astrocytes from different brain regions are equipped with metabotropic receptors that are activated by noradrenergic signaling [20,21,22,23,24,25,26,27,28,29,30,31,32,33,34,35,36,37,38].

The locus coeruleus (LC), a small nucleus located in the brainstem, is the major source of NE afferents in the central nervous system (CNS). Despite its small dimension, the LC sends complex, widespread long-range projections throughout the whole brain, including the ventral tegmental area (VTA) [39]. The VTA is a midbrain dopaminergic nucleus engaged in the processing of rewarding/aversive stimulus, motivation, reward-mediated learning, and attention [40,41,42,43]. VTA also participates in pathologies with high social costs, characterized by alterations in reward-related behaviors and by stress conditions, such as drug addiction and mood disorders [44,45]. The reciprocal interplay between neurons and astrocytes influences the output of different brain circuits, ultimately translating into a global effect in different animal behaviors such as circadian [46,47], feeding [48,49,50], goal directed [51,52,53], fear [54,55], social [56,57], and depressive [58,59,60,61] behaviors, among others. These astrocyte actions, extensively described in different brain regions in the past years, have only recently begun to be investigated in the VTA [62,63,64]. However, whether astrocytes from the VTA respond to NE has not yet been explored.

Ultrastructural studies in rodents have confirmed the presence of non-synaptic appositions between noradrenergic axons and dopaminergic dendrites in the VTA [65]. Interestingly, glia processes separated one-fourth of these non-synaptic juxtapositions. Furthermore, in VTA rodents, the α1ARs have been found in glial compartment [66]. These results suggest the possibility of a direct astrocyte response to NE in the VTA. In the present study, we provide a descriptive characterization of the astrocyte Ca^2+^ response to NE in the VTA, using the genetically encoded Ca^2+^ indicator GCaMP6f. Given that stress-related mental disorders, such as anxiety and depression, are more common in women than in men [67], we explored the astrocyte response to NE in both female and male mice. Our results show that NE activates mainly α1ARs, eliciting Ca^2+^ elevations in the soma and processes of astrocytes from the VTA. In the IP3R2^−/−^ mice, our results reveal that the residual Ca^2+^ transients still observed in the absence of the IP3R2 are mediated by the release of Ca^2+^ from the ER, probably through the activation of IP3 receptors other than IP3R2.

## 2. Materials and Methods

### 2.1. Animals

Animal care, handling, and procedures were carried out in accordance with National (D.L. n.26, 4 March 2014) and European Community Council (2010/63/UE) laws, policies, and guidelines, and they were approved by the local veterinary service. We performed experiments in inositol 1,4,5-triphosphate-type 2 receptor (IP3R2) knockout mice (IP3R2^−/−^) and IP3R2^+/+^ littermates, obtained after crossing IP3R2^+/−^ mice born from C57BL/6J wt x IP3R2^−/−^ [68] crossbreedings. PCR was used to genotype mice [63].

### 2.2. AAV Delivery

We used graduated glass pipettes to bilaterally inject the viral vector AAV5.GfaABC1D.cytoGCaMP6f.SV40 (Addgene, Watertown, MA, USA, 1.81 × 10^13^ genome copies/mL, pZac2.1 gfaABC1D-cyto-GCaMP6f was a gift from Baljit Khakh [69]), containing the astrocytic promoter GfaABC1D, to selectively express in astrocytes the genetically encoded Ca^2+^ indicator GCaMP6f, in the VTA of IP3R2^+/+^ and IP3R2^−/−^ mice, at postnatal days 28–30. For injections, animals were anesthetized with isoflurane (induction 4–5%, maintenance 1–2%). Depth of anesthesia was assured by monitoring respiration rate, eyelid reflex, vibrissae movements, and reactions to pinching the tail and toe. After drilling two holes into the skull over the VTA (coordinates from Bregma (in mm): AP −3.0, ML ± 0.5, DV −4.4), we bilaterally injected a total volume of 1 μL, without previous dilution, of AAV5.GfaABC1D.cytoGCaMP6f.SV40 per hole by using a pulled glass pipette connected to a custom-made pressure injection system.

### 2.3. Brain Slice Preparation

Three weeks after the intracranial injection, we prepared brain slices from young adult mice (around P50). Horizontal VTA slices (240 µm) were obtained from both male and female mice. Given that results obtained from different sexes were comparable, data from female and male mice were pooled together in sex-balanced groups. For slice preparations, animals were anesthetized with isofluorane, and the brain was removed and transferred into an ice-cold artificial cerebrospinal fluid (ACSF) containing (in mM) 125 NaCl, 2 KCl, 2 CaCl_2_, 1 MgCl_2_, 25 glucose, 25 NaHCO_3_, and 1.25 NaH_2_PO_4_, pH 7.4 with 95% O_2_-5% CO_2_. Slices were cut with a vibratome (Leica Vibratome VT1000S, Mannhein, Germany) in the ice-cold solution described in Dugue at al. 2005 [70] containing (in mM) 130 KGluconate, 15 KCl, 0.2 EGTA, 20 HEPES, 25 glucose, and 2 Kynurenic acid. Slices were then transferred for 1 min into a room-temperature solution containing (in mM) 225 D-mannitol, 2.5 KCl, 1.25 NaH_2_PO_4_, 26 NaHCO_3_, 25 glucose, 0.8 CaCl_2_, and 8 MgCl_2_, with 95% O_2_–5% CO_2_. Slices were loaded with the astrocyte-specific marker Sulforhodamine 101 (SR101) (0.3 µM, Sigma Aldrich, Milano, Italy) in ACSF at 32 °C for 15 min and then maintained at room temperature for the entire experiment.

### 2.4. Ca^2+^ Imaging Experiments and Analysis

Ca^2+^ imaging experiments were performed in brain slices to assess the Ca^2+^ responses evoked in astrocytes after bath perfusion with NE. The recording solution contained (in mM): NaCl 120; KCl 2; NaH_2_PO_4_ 1; NaHCO_3_ 26; MgCl_2_ 1; CaCl_2_ 2; and glucose 10 (pH 7.4 with 95% O_2_–5% CO_2_). To minimize an indirect neuronal effect in the astrocyte response observed, due to the actions exerted by NE on neurons, we included TTX 1 μM in the recording solution. A subset of Ca^2+^ imaging experiments (Figure 1) were conducted with a confocal laser scanning microscope TCS-SP5-RS (Leica Microsystems, GmbH, Wetzlar, Germany) equipped with two solid state lasers tuned at 448 nm and 543 nm (to image GCaMP6f and SR101 fluorescence, respectively) and a 20× objective (NA, 1.0). Images (8 bit depth) were acquired at a 1 Hz frame rate for 120 s (unless otherwise stated), with time intervals of 5 min between successive recordings. After three basal recordings (basal 1–3), we performed a 3 min recording immediately after starting the bath perfusion of the slice with a low concentration of NE (0.5–1 μM, [NE]_low_). NE was then washed out for ~20 min and, during the washing of the agonist, we performed two recordings (wash 1–2). Afterwards, slices were perfused for 3 min with a high concentration of NE (5–10 μM, [NE]_high_). Image sequences were processed with ImageJ 1.49v. Regions of interest (ROIs) were drawn around the cellular somata, using the red SR101 signal as a reference. Somatic Ca^2+^ events were estimated as changes of the GCaMP6f fluorescence signal over baseline (ΔF/F_0_ = (F(t) − F_0_)/F_0_). A fluorescence increase was considered a significant event when it exceeded four times the standard deviation (SD) from the baseline (mean + 4 × SD). For each peak, we collected the following parameters: the onset (first time point with ΔF/F_0_ > mean + 4 × SD), the amplitude (maximum ΔF/F_0_ value reached), and the duration (from the first to the last time point with ΔF/F_0_ > mean + 4 × SD). When the fluorescence signal between two local peaks did not return to basal levels, we counted two peaks if the recovery of the fluorescence signal before the second local peak reached half the maximum amplitude of the first local peak. When the recovery did not reach this value, we counted two peaks only if the partial recovery of the first local peak was followed by a plateau before the kinetics of the fluorescence signals rose abruptly to shape the second local peak. For each slice challenged with NE, we assigned the onset of the NE response to the onset of the first peak detected. Astrocyte Ca^2+^ response was quantified by analyzing the percentage of active astrocytes and the frequency of Ca^2+^ peaks/astrocyte in each slice. The quantification of these parameters was reported separately for the first and second minute of the astrocyte response. The astrocyte activity in basal conditions corresponded to the mean values observed in three recordings. During washing of NE, the reported astrocyte activity corresponds to the mean values of two recordings. To calculate the percentage of active astrocytes, for each slice, we assigned the value of 100% to the number of SR101^+^ astrocytes displaying at least one Ca^2+^ transient during the first minute of the response to [NE]_high_. Next, we calculated the percentage of these astrocytes that were active in each condition. The frequency of Ca^2+^ peaks/astrocyte was reported both as a mean and as a time course in bins of 15 s. When astrocytes responded to both NE concentrations, we also reported the cumulative distributions of the amplitude and duration of the Ca^2+^ peaks observed in these astrocytes during the [NE]low and [NE]high. Calcium imaging experiments reported in Figures 2–4 and 6–8 were conducted with a two-photon laser scanning microscope (Multiphoton Imaging System, Scientifica Ltd., Uckfield, UK), equipped with a pulsed red laser (Chameleon Ultra 2, Coherent, Santa Clara, CA, USA). Power at sample was 12–14 mW. GCaMP6f and SR101 were excited at 920 nm. To identify the astrocytes that express GCaMP6f, slices were excited at 820 nm. Images (12 bit depth) were acquired with a water-immersion lens (Olympus, LUMPlan FI/IR 20×, 1.05 NA), with a field of view of 120 × 120 μm at 1.5 Hz acquisition frame rate. Recordings were of 120 s (unless otherwise stated), with time intervals of 5 min between successive recordings. After three basal recordings (basal 1–3), we performed a 3 min recording immediately after starting the bath perfusion of the slice with 10 μM NE (hereafter, 1st NE). NE was then washed out and, during the washing of the agonist, we performed two recordings (wash 1–2) unless otherwise stated. Afterwards, we performed a 3 min recording during a second perfusion with 10 μM NE (hereafter, 2nd NE). When antagonists were used, the perfusion of the antagonist was initiated during the washing of the 1st NE, except for the experiments with 2-APB or BTP-2 in IP3R2^−/−^ mice. In these experiments, the washing of the 1st NE was performed in the absence of antagonist and, only after two recordings in these conditions, we then initiated the application of the antagonist. Before the application of the 2nd NE, we performed three recordings during the antagonist perfusion. In the experiments with CPA, the application of the antagonist was initiated immediately after the 1st NE and was maintained for five recordings before application of the 2nd NE. In this way, the refilling of the ER was inhibited when the Ca^2+^ content of this intracellular store was already reduced by the 1st NE application. In the experiments performed with the two-photon microscope, to define the onset of the NE response, we drew a ROI covering the entire field of view and calculated its ΔF/F_0_. We assigned the onset of the NE response to the first time point with a ΔF/F_0_ > mean + 2 × SD. Of note, the increase in the fluorescence signal of the entire field of view was mainly entrained by the changes in the astropile fluorescence, while soma responses were observed with a delay. As shown in Figure 1, the response to 10 μM NE was severely reduced during the second min after the start of the astrocyte response. For this reason, in the experiments shown in Figures 2–4 and 6–8, we restricted our analysis to the first min of the astrocyte response. To describe the somatic astrocyte response to NE, the image sequences acquired with the two-photon microscope were analyzed similarly to the image sequences acquired with the confocal microscope. In addition, for each astrocyte that showed a somatic Ca2+ peak in response to both NE applications, we also noted the maximum value of ∆F/F0 during the NE response (hereafter, maximum amplitude). Finally, for each astrocyte that responded to NE, we also measured the area under the curve (hereafter, AUC) of the somatic fluorescence changes observed during the first minute of the response to NE. When an astrocyte responded only to the 1st NE application, and not to the 2nd one, we calculated the AUC during the 2nd NE application in a time window equivalent to that of the response observed to the 1st NE application. In IP3R2^−/−^ mice, the identification of separated peaks was not reliable (see Results) and, consequently, the frequency of somatic Ca^2+^ peaks was not quantified in these mice. To extract Ca^2+^ event dynamics at astrocyte processes from the entire field of view, we first imposed a black mask covering the soma of all astrocytes and, then, we used AQuA [71] to extract astropile events in an automated, unbiased, event-based way. Events were initially extracted with a low threshold and then filtered sequentially by area (area > 5.5 μm^2^), amplitude (area < 5.5 μm^2^ but amplitude > 10 ∆F/F_0_), and duration (area < 5.5 μm^2^ and amplitude < 10 ∆F/F_0_ but duration > 3 frames, i.e., 3.9 s). This way, events were considered, independently of their amplitude and duration, if they were large and extended for an area of astropile beyond the threshold of 5.5 μm^2^. For the spatially restricted events that do not reach this area threshold, we considered them, independently of their duration, if they showed a high amplitude exceeding the threshold of 10 ∆F/F_0_. Finally, for the spatially restricted and low amplitude events, we considered them only if they were enduring and lasted longer than 3.9 s. For each slice, we measured the frequency of the astropile events in each condition and the mean area, amplitude, and duration of the events observed during the NE response. For statistical analysis of the somatic and astropile Ca^2+^ response to NE, in the experiments with a 2nd application of NE in the absence of an antagonist, we compared the response to the 2nd NE application to the response observed to the 1st NE application (percentage of 2nd NE/1st NE). In the experiments with a 2nd application of NE in the presence of an antagonist, we compared the percentage of the 2nd NE/1st NE in the presence of an antagonist to the percentage of 2nd NE/1st NE in the absence of an antagonist. In the fluorescence images shown in Figures 1e and 2b—920 nm, we applied a Kalman filter in ImageJ to remove high gain noise from the image stack. In the fluorescence images shown in Figure 2b—820 nm, we averaged 45 images to remove noise from the image.

### 2.5. Immunohistochemistry and Cell Counting

For the evaluation of the number of GCaMP6f-expressing astrocytes and neurons, we prepared 70 μm thick brain slices of young adult animals injected with AAV5.GfaABC1D.cytoGCaMP6f.SV40 and performed double immunofluorescence. Mice were euthanized with 5% isoflurane and transcardially perfused with PBS followed by ice-cold 4% PFA in PBS. Brains were removed and postfixed overnight at 4 °C in the same fixative solution. Horizontal brain slices were obtained with a VT1000S vibratome (Leica), collected as floating sections and blocked for 1 h in the Blocking Serum (BS: 1% BSA, 2% goat serum, and 1% horse serum in PBS) and 0.3% TritonX-100. After blocking, sections were incubated (overnight at 4 °C) with primary antibodies in BS plus 0.03% Tx-100: anti-NeuN (RRID:AB_2298772, 1:200 mouse, Thermofisher_Millipore, Hampton, NH, USA, MAB377) or anti-S100β (RRID:AB_2620025, 1:300 guinea pig, Synaptic System, Göttingen, Germany, 287004) antibodies. After washing with PBS, slices were incubated for 2 h at room temperature with secondary antibodies: donkey immunoglobulins anti-mouse Alexa 556 conjugated (RRID:AB_2534012, 1:500, Thermofisher, Waltham, MA, USA, A10036) or goat anti-guinea pig Alexa 546 conjugated (RRID:AB_2534118, 1:500, Thermofisher, A11074), respectively. After secondary antibody incubation, we saturated with rabbit immunoglobulins and then performed the overnight incubation with the directly Alexa 488 conjugated rabbit polyclonal anti-GFP (RRID:AB_221477, 1:300, Thermofisher, A21311) to identify GCaMP6-expressing cells. Slices were then washed, and nuclei were stained with TOPro3 (Invitrogen Thermo-Scientific, Waltham, MA, USA, 1:1000) to identify single cells. Negative controls were performed in the absence of the primary antibodies. We used a TCS-SP5-RS laser scanning microscope (Leica, Wetzlar, Germany; 20x NA1x/W objective, pinhole value resulting in 1 μm optical thickness) to acquire sequential three channels, confocal image z-stacks (1 μm z-step, 456.33 × 456.33 μm), and ImageJ for double-labeled cell counting. Positive somata were identified in single confocal planes. To quantify the cellular specificity of GCaMP6f expression, we counted GCaMP6^+^ cells and then we evaluated the percentage of GCaMP6^+^ cells that were neurons (NeuN^+^/total GCaMP6^+^) or astrocytes (S100β^+^/total GCaMP6^+^). To quantify the efficiency of GCaMP6f expression in the neuronal and astrocytic populations, we counted neuron (NeuN^+^) or astrocyte (S100β^+^) cells and then we evaluated the percentage of neurons that were GCaMP6^+^ (GCaMP6^+^/total NeuN^+^) and the percentage of glial cells that were GCaMP6^+^ (GCaMP6^+^/total S100β^+^). VTA from both hemispheres of injected animals was evaluated in 1–2 fields of view of three mice for each group. For the evaluation of the number of GCaMP6f^+^ astrocytes that express the α1ARs, a similar approach was used with the primary antibody anti-α1AR (RRID:AB_2273801, 1:200, Thermofisher, PA1-047), revealed by goat anti-rabbit immunoglobulins Alexa Fluo 555 conjugated (RRID: AB_2535851, 1:500, Thermofisher, A21430). We acquired confocal image z-stacks (0.7–1 μm z-step, 170 × 170 μm). In ImageJ, we quantified GCaMP6f^+^ cells that display α1AR staining at soma and/or thick proximal processes, in continuity with the soma, at single confocal planes, independently of the quantitative level of staining.

### 2.6. Drugs

All drugs were bath perfused. TTX, prazosin, 2-APB, NBQX, DAP5, MPEP, CGP55845, PPADS, and ZM241385 were from HelloBio (Dunshaughlin, Ireland). RWL (rauwolscine), eticlopride, and picrotoxin were from Abcam (Cambridge, UK). BTP-2 was from Tocris (Bristol, UK). NE and CPA (cyclopiazonic acid) were from Merck (Milano, Italy).

### 2.7. Data Analysis

Data analysis was performed with Origin 8.0 (Microcal Software, Piscataway, NJ, USA), Microsoft Excel 2010, ImageJ (NHI), Sigma Plot 11, and AQuA 2020 [71]. Data are expressed as mean ± standard error of the mean (SEM) or presented as box and whisker plots in which each box was defined by the 25th and 75th percentiles, where the central line indicated the median, the red line indicated the mean value, the whiskers represented the 10th and 90th percentiles, and gray circles the 5th and 95th percentiles. In box and whisker plots, individual percentage values are shown as circles on the right of each box, and the thickness of the circle line correlates with the number of repetitions of that value. A normality test was applied to the data before running statistical tests. Based on the normality test result, data were analyzed using either parametric or nonparametric tests, as appropriate. Tests used are reported in figure legends. For cumulative distribution comparisons, we applied the Kolmogorov–Smirnov test. Statistical differences: *p* < 0.05 (*), *p* < 0.01 (**), *p* < 0.001 (***), not significant (ns).

## 3. Results

### 3.1. NE Triggered Somatic Ca^2+^ Increases in VTA Astrocytes

The goal of our study was to investigate whether NE triggers Ca^2+^ transients in VTA astrocytes. Wild-type mice (hereafter, IP3R2^+/+^ mice) were injected in the VTA with an AAV to express GCaMP6f specifically in astrocytes (Figure 1a). The specificity of the expression of GCaMP6f in astrocytes was verified using immunohistochemistry tools (Figure 1b,c).

Fresh VTA slices were loaded with the astrocyte-specific marker sulforhodamine 101 (SR101), and changes in the fluorescence signal of GCaMP6f were detected with a confocal microscope in SR101^+^ cells. The confocal Ca^2+^-imaging protocol used is depicted in Figure 1d. As shown in Figure 1e–g, in the presence of TTX, astrocytes from the VTA responded with somatic Ca^2+^ transients to both low and high concentrations of NE. For each slice, we assigned the value of 100% to the number of SR101^+^ astrocytes displaying at least one Ca^2+^ transient during the first minute of the response to [NE]_high_. Then, we calculated the percentage of active astrocytes per min in each condition. This value significantly decreased during the second minute of [NE]_high_ application (Figure 1h). Nonetheless, this value did not return completely to the levels observed during basal and wash conditions. The percentage of astrocytes displaying Ca^2+^ transients during the first minute of the response to [NE]_low_ showed a tendency to be lower compared to the percentage of active astrocytes observed in response to the [NE]_high_. Interestingly, the percentage of active astrocytes did not show a significant reduction during the second minute of [NE]_low_. Furthermore, during the second minute of the astrocyte response to NE, the frequency of Ca^2+^ peaks per astrocyte was significantly reduced only when astrocytes were challenged with a high concentration of NE; nonetheless, the frequency during the first minute tended to be higher in response to [NE]_high_ compared to the response observed to [NE]_low_ (Figure 1j). These results suggest that the astrocyte response to the bath perfusion with NE exhibited an oscillatory behavior that lasts longer with lower concentrations of NE (see also Figure 1f). However, when astrocytes responded to both NE concentrations, the astrocyte response to [NE]_high_ was more robust in terms of the amplitude and the duration of the Ca^2+^ transients observed (Figure 1k; mean amplitude: [NE]_low_, 2.07 ± 0.12 ∆F/F_0_ vs. [NE]_high_ 2.72 ± 0.14 ∆F/F_0_; mean duration: [NE]_low_, 13.2 ± 1.33 s vs. [NE]_high_ 23.45 ± 1.95 s).

In other brain circuits, the astrocyte response to NE has been investigated by challenging astrocytes with [NE] around 10 μM or higher [22,27,36,72]. To leave the option to compare the astrocyte responses to NE in the VTA and in other brain regions and given that the VTA astrocyte response to [NE]_high_ is more robust, we performed the following experiments using 10 μM NE. As previously reported, the response to 10 μM NE was severely reduced during the second minute after the start of the astrocyte response, and therefore, in the following experiments, we restricted our analysis to the first minute of the response.

### 3.2. Astrocyte Response to NE at Soma and Processes Was Similar in Female and Male Mice

The use of the GCaMP6 Ca^2+^ indicators and advanced microscopy allows studying the astrocyte Ca^2+^ signaling at the level of fine processes with unprecedented detail [9]. To further explore the VTA astrocyte response to NE at both soma and fine processes, we next moved to two-photon microscope experiments (Figure 2a). With the low cytoplasmic Ca^2+^ levels present at basal conditions, the fluorescence signal of GCaMP6f was barely detectable when slices were excited at 920 nm (the λ_excitation_ at which the GCaMP6f fluorescence signal increased upon the binding to Ca^2+^, Figure 2b, right). In contrast, at 820 nm (the isosbestic GCaMP6 λ_excitation_), the fluorescence signal of the indicator did not depend on the binding to Ca^2+^ and it was easily detected at the soma of astrocytes expressing GCaMP6f (Figure 2b, left). Thus, we used the excitation at 820 nm to identify all astrocytes expressing GCaMP6f, and then we investigated the response of GCaMP6f^+^ astrocytes during the challenging with NE at 920 nm.

Figure 2c shows that, in the presence of TTX, almost all astrocytes expressing GCaMP6f responded with somatic Ca^2+^ transients to NE during the first minute of the response. When we split by sex the data collected from slices of female and male mice, we found no differences between sexes in the percentage of activated somata (Figure 2c), nor in the frequency of somatic Ca^2+^ peaks per astrocyte evoked during the NE challenging (Figure 2d). Furthermore, for each astrocyte that responded to NE, analysis of the maximum value of ∆F/F_0_ (hereafter, maximum amplitude) and of the area under the curve (hereafter, AUC) of the somatic fluorescence changes observed during the first minute of the response to NE revealed no differences in slices from female and male mice (maximum amplitude: NE female, 13.38 ± 0.57 ∆F/F_0_ vs. NE male, 13.68 ± 0.67 ∆F/F_0_, ns; AUC: NE female, 280.96 ± 15.03 arbitrary units (a.u.) vs. NE male, 279.76 ± 15.83 (a.u), ns; two-sample *t* test, *n* = 71 and 70 astrocytes, respectively). After analyzing the somatic Ca^2+^ response, we next used the open-source AQuA [71], a software that works in an automated and event-based way, to extract the Ca^2+^ events at astrocyte processes from the entire field of view (astropile; Figure 2e). Figure 2e,f show that, compared to the frequency observed at basal conditions, the number of Ca^2+^ events at astropile increased robustly during the first minute of the astrocyte response to NE (astropile Ca^2+^ events/min: NE, 1320 ± 123 vs. basal, 93 ± 12; *p* < 0.001). Similarly to the response at the soma, we observed no differences in the astropile response between female and male mice in terms of astropile Ca^2+^ events/min (Figure 2f). The analysis of the mean area, amplitude, and duration of the Ca^2+^ events detected at astropile for each slice challenged with NE revealed no differences between female and male responses (mean area: NE female, 4.16 ± 0.21 μm^2^ vs. NE male, 4.39 ± 0.45 μm^2^; ns; mean amplitude: NE female, 13.61 ± 0.7 ∆F/F_0_ vs. NE male, 14.46 ± 0.87 ∆F/F_0_; ns; mean duration: NE female, 2.61 ± 0.05 s vs. NE male, 2.55 ± 0.08 s, ns; Mann–Whitney rank sum test, *n* = 17 and 18 slices, respectively). Given that astrocytes from mice of both sexes showed similar responses to NE, in the following experiments, we pooled data obtained in sex-balanced experimental groups.

### 3.3. α1 ARs Mediated VTA Astrocyte Ca^2+^ Responses to NE

To investigate the adrenergic receptor that mediates the VTA astrocyte response to NE, we performed a second challenge with NE in the presence of antagonists for different adrenergic receptors. First, to evaluate the response to multiple applications of NE in the same slice, in a subset of the experiments shown in Figure 2, we performed a second challenge with NE in the absence of antagonists (Figure 2a). As shown in the heatmaps reporting the dynamics of the fluorescence changes in the soma of all GCaMP6f^+^ astrocytes investigated, both the first and second applications of NE triggered a robust astrocyte response that consisted of oscillatory Ca^2+^ increases in most of the responsive astrocytes (Figure 3a). When we plotted the number of somatic Ca^2+^ peaks per astrocyte, we observed a strong increase in the Ca^2+^ signaling of astrocytes also upon the second NE application (Figure 3b). Compared to the 1st NE, the 2nd NE challenge activated the same number of astrocytes, and no significant differences were found in the frequency of Ca^2+^ peaks per astrocyte, the maximum amplitude of the response, and the AUC of the ∆F/F_0_ signal, although the first two parameters showed a tendency to diminish during the 2nd NE response (Figure 3c). Analysis of the Ca^2+^ response at the astrocyte processes revealed that the number of Ca^2+^ events observed upon the 2nd NE application was significantly reduced compared to the 1st NE application, although approximately a 75% of the response to the 1st NE was preserved during the 2nd NE application (Figure 3d,e). Analysis of the different parameters that describe the Ca^2+^ events observed at the astropile revealed that only the mean area showed a small but significant rundown during the 2nd NE application (Figure 3e). These results suggest that the cellular machinery at the basis of the NE-evoked Ca^2+^ transients was roughly restored during the wash phase of our Ca^2+^-imaging protocol, and the Ca^2+^ response evoked by the 2nd NE application showed only a slight rundown.

To further isolate the astrocyte response from the indirect NE effects on neurons, we performed experiments in the presence of TTX and antagonists for the following receptors: GABA_A_ (Picrotoxin 50 μM), GABA_B_ (CGP55845 5 μM), AMPARs (NBQX 10 μM), NMDARs (DAP5 50 μM), mGluR5 (MPEP 20 μM), purinergic P2Rs (PPADS 25 μM), and adenosine A2ARs (ZM241385 1 μM). As shown in Appendix A, most of the parameters of the astrocyte response to NE were preserved in the presence of this mix of blockers, supporting the hypothesis of a direct response of VTA astrocytes to NE.

In the presence of the antagonist prazosin, which inhibits mainly the α1ARs and, with a lower affinity, the α2ARs, the somatic astrocyte response to NE was almost completely abolished (Figure 4a). The number of active astrocytes, the frequency of Ca^2+^ peaks, and the AUC during the 2nd NE were drastically reduced compared to the 1st NE application (Figure 4d). Further, the mean maximum amplitude of the few somatic Ca^2+^ transients still observed in the presence of prazosin, that were detected at a low frequency similar to that displayed at basal conditions, showed a significant reduction compared to the maximum amplitude of the peaks observed in the group without an antagonist (Figure 4d). In contrast, in the presence of the selective α2AR antagonist RWL, the astrocyte response to NE was highly preserved (Figure 4b) and, compared to the group without an antagonist, only a significant reduction was detected in the AUC of the fluorescence trace (Figure 4d). Given that the D2-type dopamine receptors can be activated by high concentrations of NE [73], we also evaluated the role of these receptors in the astrocyte response to NE in the VTA. In the presence of the D2-type dopamine receptor antagonist eticlopride, the astrocyte response to NE was well preserved (Figure 4c), and only the AUC during the stimulation with NE showed a significant reduction (Figure 4d). The analysis of the frequency of the Ca^2+^ events at processes revealed an impairment of the astropile response to NE in the presence of prazosin, while RWL and eticlopride induced only a small reduction that did not reach statistical significance (Figure 4e–h). Further analysis of the different parameters that describe the Ca^2+^ events observed at the astropile revealed that only the mean area showed a significant reduction during the 2nd NE application in the presence of prazosin (Figure 4h). In summary, although these data do not allow us to completely exclude a minor, modulatory role for α2ARs and D2-type dopamine receptors, our results suggest that the α1ARs are the main mediators of the astrocyte response evoked by NE in the VTA (Figure 4i).

To provide structural evidence of the presence of α1ARs in mouse VTA astrocytes expressing GCaMP6f, we next performed immunohistochemistry staining experiments. Our results revealed the presence of a high level of punctate staining for α1ARs in mouse VTA. This staining occurred at the level of the soma of cells not expressing GCaMP6f that, according to their large size, were probably dopaminergic neuronal cells (Figure 5a). Interestingly, the α1AR staining was also observed at the soma of astrocytes expressing GCaMP6f (Figure 5a,b). In astrocyte soma, the detected expression levels were highly variable, with some cells displaying high levels (upper panels of Figure 5b) and others exhibiting low levels (lower panels of Figure 5b). Likewise, α1ARs were also observed in thick processes in continuity with an astrocyte soma (Figure 5b), sometimes even in the absence of soma staining. We quantified the percentage of GCaMP6f^+^ astrocytes expressing α1ARs and we found that around 58% of GCaMP6f^+^ astrocytes were positive for the expression of α1ARs in the VTA (Figure 5c). Remarkably, a high level of staining was observed also in regions excluding soma compartments (Figure 5a,b).

### 3.4. Astrocytes from IP3R2^−/−^ Mice Showed a Reduced Response to NE

The α1ARs are Gq-protein-coupled receptors (GqCPRs) linked to the production of IP3. To gain further insights into the mechanism of VTA astrocyte response to NE, we thus performed experiments in slices from IP3R2^−/−^ mice in which Ca^2+^ elevations mediated by GPCRs are largely impaired [33,72,74]. In VTA slices from these mice, we observed that NE application still evoked a significant Ca^2+^ response at both soma and processes (Figure 6a,d), with no differences between female and male astrocyte responses (Figure 6a,d). Similarly, in slices from female and male IP3R2^−/−^ mice, we observed no differences in the maximum amplitude of the somatic Ca^2+^ responses evoked, and we found only a significant reduction in the AUC in male astrocytes (maximum amplitude: NE female, 4.70 ± 0.31 ∆F/F_0_ vs. NE male, 4.93 ± 0.47 ∆F/F_0_, ns; AUC: NE female, 135.15 ± 8.93 a.u. vs. NE male, 107.37 ± 10.04 a.u., *p* < 0.05; Mann–Whitney rank sum test, *n* = 71 and 61 astrocytes, respectively). Finally, analysis of the mean area, amplitude, and duration of the Ca^2+^ events detected at the astropile of all slices challenged with NE revealed no differences between female and male NE responses in IP3R2^−/−^ mice (mean area: NE female, 3.05 ± 0.17 μm^2^ vs. NE male, 2.93 ± 0.15 μm^2^; ns; mean amplitude: NE female, 14.53 ± 0.6 ∆F/F_0_ vs. NE male, 15.49 ± 0.75 ∆F/F_0_; ns; mean duration: NE female, 2.41 ± 0.07 s vs. NE male, 2.35 ± 0.08 s, ns; Mann–Whitney rank sum test: area, amplitude; two-tailed *t* test: duration; *n* = 19 and 18 slices, respectively). In summary, these results suggest that the response to NE observed in IP3R2^−/−^ mice is similar in female and male mice. Accordingly, in IP3R2^−/−^ mouse experiments, we pooled data obtained in sex-balanced experimental groups.

As expected, when we compared the response to NE in astrocytes from IP3R2^−/−^ and IP3R2^+/+^ mice, we observed a slight, although significant, reduction in the number of responsive astrocytes to NE in IP3R2^−/−^ mice (Figure 6b,c). In IP3R2^−/−^-responsive astrocytes, the small increase in the GCaMP6f fluorescence signal at the soma was relatively sustained over time (Figure 6b, note that the identification of separated peaks was not reliable in these mice) and, compared to the values observed in IP3R2^+/+^ astrocytes, it reached a lower maximum amplitude and showed a diminished integral of the curve (Figure 6c). Furthermore, at the level of astropile, IP3R2^−/−^ mice showed a reduction in the number of Ca^2+^ events detected as well as a significant decrease in the area and duration of these Ca^2+^ events that show, however, a similar amplitude (Figure 6e,f). These results agree with the extensive literature that highlights the pivotal role of IP3R2 in astrocyte Ca^2+^ signaling and the presence of a residual astrocyte response, at least in adult mice, in the absence of the type-2 receptor for the IP3 (see Section 4).

### 3.5. NE-Triggered Astrocyte Response in IP3R2^−/−^ Mice Depended on ER Intracellular Ca^2+^ Stores and IP3Rs

In IP3R2^−/−^ mice, intracellular and/or extracellular sources have been indicated as possible contributors to the shaping of Ca^2+^ transients in astrocytes from these mice [9]. To evaluate the role of the endoplasmic reticulum (ER) Ca^2+^ in the astrocyte response to NE that persist in IP3R2^−/−^ mice, we used a pharmacological approach. In a subset of the experiments reported in Figure 6, we first performed a second challenge with NE in the absence of antagonists to assess the response to several applications of NE in slices from IP3R2^−/−^ mice. As observed in IP3R2^+/+^ mice, also in IP3R2^−/−^ mice, the astrocyte response triggered by a 2nd NE application was similar to that evoked by the 1st NE application (Figure 7a). The number of active astrocytes, the maximum Ca^2+^ peak amplitude, and the AUC of the Ca^2+^ signal during the first minute of the response to NE were not significantly reduced during the 2nd NE application compared to the 1st NE challenging (Figure 7b). In agreement with these results, analysis of the Ca^2+^ response at the astrocyte processes revealed a similar frequency, mean area, mean amplitude, and mean duration of the Ca^2+^ elevations in the astropile during the 1st and 2nd NE applications (Figure 7c,d).

To investigate the role of other IP3Rs different from IP3R2 (i.e., IP3R1 or IP3R3) in the residual astrocyte response to NE in the VTA of IP3R2^−/−^ mice, we applied NE in the presence of 2-APB, an antagonist for all IP3R subtypes. We found that, when the IP3R signaling was blocked, the response to NE was completely impaired (Figure 8a). The number of astrocytes displaying somatic Ca^2+^ transients during NE application in the presence of 2-APB was almost zero, and, consequently, also the mean AUC was around zero (Figure 8d). The IP3R antagonist 2-APB also inhibited, with a low affinity, the Ca^2+^ release-activated Ca^2+^ channels (CRAC channels), important in the mechanism of the store-operated Ca^2+^ entry (SOCE). To evaluate the role of SOCE in the response observed in IP3R2^−/−^ astrocytes, we inhibited CRAC channels with the selective SOCE antagonist BTP-2. As shown in Figure 8b, the astrocyte response to NE was highly preserved in the presence of BTP-2 in terms of number of active astrocytes, maximum amplitude of Ca^2+^ transients, and integral of the astrocyte response (Figure 8d). These results suggest that 2-APB inhibits the NE response mainly through the inhibition of the IP3R1 or IP3R3 receptors and that the ER Ca^2+^ plays a pivotal role in the Ca^2+^ response evoked by NE in IP3R2^−/−^ mice. To confirm the role played by Ca^2+^ from this intracellular store, after the first application of NE, we immediately inhibited the refilling of the ER with cyclopiazonic acid (CPA), an antagonist of the ER Ca^2+^-ATPase (SERCA) pump. We found that, in the slices treated with CPA, VTA astrocytes did not respond to NE (Figure 8c,d). In support of the central role of Ca^2+^ from the ER in IP3R2^−/−^ mice, analysis of the Ca^2+^ events at astrocyte processes showed that the frequency and mean area of the astropile Ca^2+^ responses to NE were also negatively affected by 2-APB or CPA, while they were highly preserved in the presence of BTP-2 (Figure 8e–h). Furthermore, the few astropile Ca^2+^ events observed during the 2nd NE challenging showed a reduced duration in the presence of 2-APB and a reduced amplitude upon CPA treatment. In summary, our results suggest that the astrocyte Ca^2+^ response evoked by NE in the VTA of IP3R2^−/−^ mice depends on the Ca^2+^ extrusion from the ER intracellular store through IP3Rs other than IP3R2 (Figure 8i).

## 4. Discussion

In our study in acute mouse brain slices, we show that the agonist NE is a potent stimulus to elicit Ca^2+^ elevations in astrocytes from the VTA of mature animals, at least during young adulthood (PN50). In hippocampal and cortical astrocytes, a previous study showed that, compared to the Ca^2+^ responses evoked by acetylcholine or agonists for PAR1 and metabotropic mGluR5 receptors, the response to ATP was the astrocyte response better preserved during aging [75]. In our VTA slices, we challenged astrocytes with 100 μM ATP or 10 μM NE and the Ca^2+^ activity in these cells was drastically lower during the ATP challenge (Appendix A). These results suggest that VTA astrocytes from young adult mice exhibit a high responsiveness to NE, compared to ATP. Another feature of the NE response observed in our study is that it can be similarly elicited in astrocytes from both female and male mice, ruling out the existence of marked differences between sexes in the astrocyte response to noradrenergic signals in the circuit of the VTA. In our experiments, we applied NE in bath perfusion. This type of agonist application, which lasted several minutes before washing of the NE, evoked in astrocytic soma responses that were oscillatory with the two concentrations of NE tested. The most straightforward interpretation of the Ca^2+^ fluctuations seen during NE challenging would be the continuous presence of the agonist in the extracellular space along with the inhibitory control that the high concentrations of Ca^2+^ exert on the channel activity of the IP3Rs [76]. Interestingly, in spite of the strong response triggered by 10 μM NE, in terms of amplitude of the response and duration of the peaks, an oscillatory dynamic in the response is still detected, suggesting that the level of desensitization of the astrocyte response to NE is not high. In olfactory bulb circuits, low concentrations of NE (3 μM) evoke weak, if present, astrocyte responses [22]. In contrast, astrocytes from neocortex show a higher sensitivity to NE, displaying Ca^2+^ elevations in response to NE challenge when the [NE] is in the sub-μM range [32]. In our experiments, we observed consistent astrocyte Ca^2+^ elevations triggered by NE in the range of 0.5–1 μM. These results highlight the remarkable sensitivity that VTA astrocytes seem to display to NE. Considering the high sensitivity and responsiveness that VTA astrocyte display to NE, along with the low desensitization that the response seems to exhibit, it is tempting to speculate that astrocytes may have a substantial impact in the effects that NE exerts in VTA circuits.

Our functional results show that VTA astrocytes respond to NE mainly through α1ARs. Our immunohistochemistry experiments also provide anatomical evidence of the presence of α1ARs in mouse VTA astrocytes, at least at the level of the soma and thick proximal processes. These results expand previous studies illustrating the presence of α1ARs in rat VTA glial cells [66]. The use of state-of-the-art techniques to specifically downregulate α1ARs in astrocytes would be valuable in providing additional evidence for the functional presence of these receptors in VTA astrocytes. We also detected the presence of α1ARs at the soma of neurons and, at high levels, at regions not associated with cellular somata. Although the spatial resolution of the confocal microscopy does not allow us to unambiguously ascribe this staining to fine neuronal or astrocyte processes, it is probable that this staining covers both neuropile and astropile compartments. Of note, while our anatomical data reveal the presence of α1ARs in a relatively high percentage, but not all, of astrocyte somata (around 58%), our functional results show an astrocyte response at the soma in almost all analyzed astrocytes. In astrocytes lacking the adrenergic receptor at the soma, the astrocyte response at the somatic compartment may occur by integration of Ca^2+^ elevations at different fine processes that travels as an intracellular Ca^2+^ wave to reach the soma of the cell. Alternatively, the Ca^2+^ response in these astrocytes could occur through the spread of an intracellular signal among neighboring cells connected by gap-junctions [77]. In our study, we did not investigate whether NE elicits cytosolic cAMP elevations in VTA astrocytes through the activation of βARs, limiting our research to Ca^2+^ signaling. Additional experiments are necessary to specifically address the activation of this important signaling pathway in VTA astrocytes during noradrenergic challenges [78].

Given the fact that the study of the neuron–astrocyte interactions struggles with the limitation that, too often, both neurons and astrocytes express the same receptors for neurotransmitter on their plasma membranes, it is likely that NE effects in VTA circuits, previously ascribed to direct effects on neurons, could be mediated, at least in part, by the recruitment of astrocytes. For example, in VTA and other brain circuits, astrocytes exert significant modulatory actions in the neuronal synaptic transmission [63,79,80]. At the same time, the activation of α1ARs in the VTA seems to modify both the excitatory and the inhibitory synaptic transmission onto VTA dopaminergic neurons [81,82]. Thereby, it would be of interest to investigate whether astrocytes participate in the mechanism of modulation of the synaptic transmission in the VTA mediated by the noradrenergic signaling.

The α1ARs, which are the main mediators of the astrocyte response to NE in the VTA, exhibit an intermediate affinity for NE compared to α2ARs and βARs (α2ARs~50 nM, high affinity > α1ARs~300 nM, intermediate affinity > β1-3ARs~800 nM, low affinity) [83]. According to the differential affinities exhibited by the several adrenergic receptors, it has been hypothesized that the activation of the different receptors could reliably follow the diverse firing patterns of LC neurons (i.e., low or high tonic and phasic neuronal discharges), which are associated with different behavioral states and build up different extracellular concentrations of NE [83,84]. According to this view, it has been hypothesized that α1AR activation could be linked to the phasic firing of LC neurons during active wake states, characterized by a significant interaction with biologically salient stimuli, as well as to the high-tonic firing associated to stressful conditions [83,85]. If in these contexts the noradrenergic afferents to the VTA are active and efficiently light up astrocytes, these glial cells may emerge as new players in the mechanisms that mediate the aforementioned behavioral states.

Previous studies investigating the role of the IP3R2 in the astrocyte Ca^2+^ response to several neurotransmitters showed that, in different brain regions from adult mice, the deletion of the IP3R2 almost completely abolishes the response at the soma of astrocytes while the Ca^2+^ response at fine processes are only partially attenuated [33,36,38,56,86,87]. In accordance with these studies, in our experiments conducted in astrocytes from adult IP3R2^–/–^ mice, we observed a Ca^2+^ response to NE that, even if reduced, is still rich in VTA astrocytes. In contrast to other brain regions, however, the response to NE is not spatially limited to the fine processes. Instead, an unexpected response is still observed at the somatic compartment. Although the reasons for these differences are not known, we hypothesize that the mechanisms at the basis of the high responsiveness and sensitivity to NE that VTA astrocytes display in IP3R2^+/+^ mice might also explain the largely preserved NE response that we observe in VTA astrocytes from IP3R2^−/−^ mice. In IP3R2^−/−^ mice, intracellular and/or extracellular sources have been indicated as possible Ca^2+^ sources shaping Ca^2+^ transients in astrocytes from these mice [9]. Along with the mitochondria [72], the ER is an intracellular store that releases Ca^2+^ even in the absence of IP3R2 receptors [36], possibly through IP3R1/3 and/or ryanodine receptors. In our experiments with the IP3R antagonist 2-APB, we observed an impairment of the residual response triggered by NE in IP3R2^−/−^ astrocytes, supporting the hypothesis that the IP3R2 is not the sole functional IP3R subtype in adult astrocytes [10], also in VTA circuits (Figure 8). In astrocyte–neuron junctions from the cortex, an elegant study found that IP3R1 is among the proteins for which the gene expression is enriched in astrocytes, suggesting that IP3Rs other than IP3R2 participate to the biology of astrocytes [88]. In line with this hypothesis, a previous study showed a role for IP3R1 in the Ca^2+^ activity and reactiveness of astrocytes from the spinal cord during IL6-mediated, STAT3 astrocyte activation necessary for chronic itch [89]. In this study, the authors demonstrated that the use of shRNAs for knocking down specifically in astrocytes the IP3R1, but not the IP3R2, ameliorates the skin inflammation observed during chronic itch, providing evidence for the existence of different roles for each IP3R subtype expressed in astrocytes. Interestingly, the IP3R1-mediated Ca^2+^ response and the subsequent effects mediated by IP3R1 activation depend on a SOCE-mediated Ca^2+^ influx from the extracellular space through the TRPC3 channels. In our experiments in the presence of BTP-2, an antagonist for the SOCE mechanism, we observed no effects of SOCE blockade on the IP3R-dependent Ca^2+^ response in IP3R2^−/−^ astrocytes. In olfactory bulbs, astrocytes respond to NE with a sustained Ca^2+^ elevation in which the late plateau-like phase depends on an Orai1-mediated SOCE mechanism while the initial Ca^2+^ peak is mainly mediated by α1ARs and α2ARs [22]. In our study in IP3R2^+/+^ astrocytes, we did not assess the contribution of SOCE mechanisms to the astrocyte response to NE in the VTA, and we cannot exclude that a cooperation exist between intracellular and extracellular Ca^2+^ sources to shape the Ca^2+^ responses observed in our experiments.

## 5. Conclusions

Our results illustrate how NE triggers Ca^2+^ transients in astrocytes, mainly through the activation of α1ARs, in the circuit of the VTA, and they invite to re-examine the interpretations previously advanced for the comprehension of the cellular and molecular mechanisms at the basis of the NE effects in this neuronal circuit. The IP3R2^−/−^ mouse model has been useful to investigate the role of astrocyte Ca^2+^ signaling in different brain circuits and animal behaviors [56,90,91,92], including those modulated by NE signaling [33,35]. However, the significant residual response to NE present in IP3R2^−/−^ VTA astrocytes raises non-negligible concerns about the use of this mouse model for the study of the role played by astrocytes in mouse behaviors dictated by the noradrenergic signaling in VTA circuits. Although a partial attenuation of the astrocyte NE response could still yield valuable results, progress in this direction would be better achieved by using state-of–the–art techniques that allow manipulating the expression level of receptors specifically in astrocytes.

## Figures and Tables

**Figure 1 cells-14-00024-f001:**
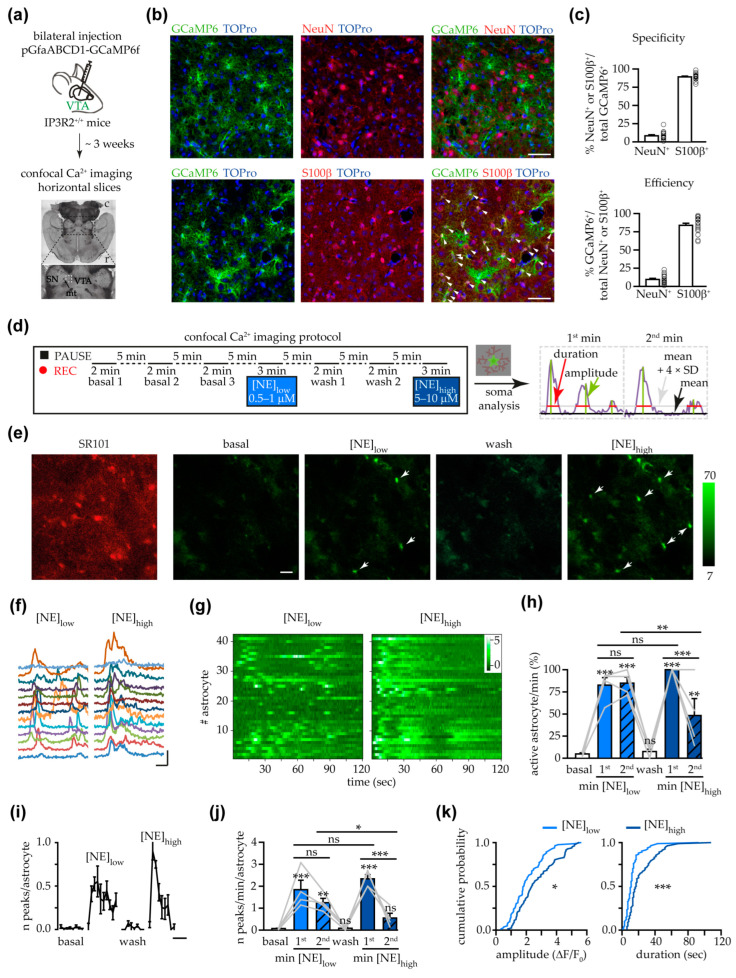
NE triggered somatic Ca^2+^ increases in VTA astrocytes. (**a**) Schematic of the experimental design for the expression of GCaMP6f in VTA astrocytes and preparation of horizontal brain slices, containing the lateral part of the posterior VTA (dashed white region in the higher magnification image), for confocal Ca^2+^ imaging experiments. c, caudal; r, rostral; VTA, ventral tegmental area; SN, substantia nigra; mt, medial terminal nucleus of the accessory optical tract. (**b**) Confocal images showing the green fluorescence of GCaMP6f (α-GFP), nuclear TOPro3 (blue), and the specific red staining for either neurons (α-NeuN) or astrocytes (α-S100β). White arrowheads, S100β^+^ GCaMP6f^+^ cells. Scale bar, 50 µm. (**c**) Specificity, bar chart showing the percentage of GCaMP6f^+^ cells that are neurons (NeuN^+^ clls) or astrocytes (S100β^+^ cells). αNeuN, *n*  =  1806 GCaMP6f^+^ cells from three mice, 8 slices, 15 fields; αS100β, *n*  =  2219 GCaMP6f^+^ cells from three mice, 6 slices, 16 fields. Efficiency, bar chart showing the percentage of neurons (NeuN^+^ cells) or astrocytes (S100β^+^ cells) that are GCaMP6f^+^. αNeuN, *n*  =  1557 NeuN^+^ cells from three mice, 8 slices, 15 fields; αS100β; *n*  =  2397 S100β ^+^ cells from three mice, 6 slices, 16 fields. (**d**) Left, schematic of the confocal Ca^2+^ imaging protocol. Right, representative example of a somatic Ca^2+^ trace showing the threshold for the identification of peaks and the parameters quantified. (**e**) Left panel, fluorescence image of SR101^+^ astrocytes from a representative experiment. Right panels, images (after Kalman filter) of the same field shown on the left, illustrating the GCaMP6f fluorescence signals at the different experimental conditions. Arrows, SR101^+^ astrocytes displaying Ca^2+^ elevations during NE challenge. Scale bar, 25 µm. (**f**) Examples of somatic Ca^2+^ dynamics (randomly colored) observed after stimulation with low or high [NE] of the same population of VTA astrocytes. Scale bars, 2 ∆F/F_0_; 20 s. (**g**) Heatmaps showing the ∆F/F_0_ of the soma of all astrocytes analyzed during the challenging with low and high [NE] (*n* = 42 astrocytes from 4 slices, 3 mice). (**h**) Bar chart showing the percentage of active astrocytes/min under different conditions (*n* = 4 slices with 42 astrocytes from 3 mice; one-way RM ANOVA and Holm–Sidak post-test). Grey lines in this and other figures with repeated measures, individual values for each slice. For each condition, it is shown above each bar the statistical analysis referring to the basal condition. The statistical analysis of the most relevant pairwise comparisons are shown above horizontal lines. (**i**) Time course of the mean number of somatic Ca^2+^ peaks per astrocyte under different experimental conditions. Scale bar, 1 min. (**j**) Bar chart displaying the mean number of somatic Ca^2+^ peaks per min and per astrocyte under different experimental conditions (*n* = 4 slices with 42 astrocytes from 3 mice; one-way RM ANOVA and Holm–Sidak post-test). (**k**) Cumulative probabilities of the amplitude and the duration of the peaks triggered by low and high [NE] in astrocytes responsive to both NE concentrations ([NE]_low_, *n* = 83 peaks; [NE]_high_, *n* = 82 peaks; Kolmogorov–Smirnov test). *p* < 0.05 (*), *p* < 0.01 (**), *p* < 0.001 (***), not significant (ns).

**Figure 2 cells-14-00024-f002:**
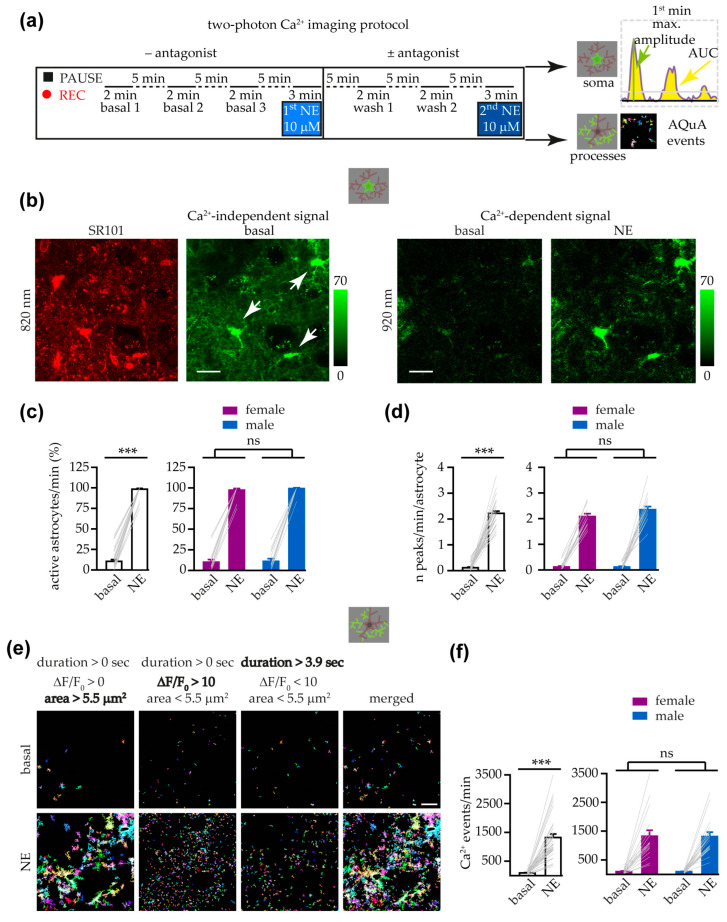
Astrocyte response to NE at soma and processes was similar in both female and male mice. (**a**) Left, schematic of the two-photon Ca^2+^ imaging protocol. NE was bath applied at 10 μM, first in the absence of an antagonist (1st NE, this figure) and, a second time, either in the absence or presence of an antagonist (following figures). Right upper panel, representative example of a somatic Ca^2+^ trace showing the threshold for the identification of peaks and the parameters quantified. Right lower panel, examples of Ca^2+^ events extracted by AQuA in the astropile. (**b**) Left panels, averaged images of SR101^+^ cells and corresponding GCaMP6f fluorescence levels at 820 nm. Arrows, SR101^+^ astrocytes expressing GCaMP6f. Right panels, fluorescence images (after Kalman filter) showing the GCaMP6f fluorescence at 920 nm of the astrocytes indicated on the left. Scale bar, 20 µm. (**c**) Bar charts showing the percentage of active astrocytes/min, at basal conditions and during the 1st NE application, in all the slices investigated (white bars, *n* = 35 slices with 144 astrocytes from 20 mice; Wilcoxon signed rank test) and in slices from only female (*n* = 17 slices with 73 astrocytes from 9 mice) or male mice (*n* = 18 slices with 71 astrocytes from 11 mice) (Aligned Rank Transformation two-way RM ANOVA with Bonferroni’s multiple comparison test; sex, ns; treatment, *p* < 0.001; no interaction). (**d**) Same as in (**c**) for the frequency of somatic Ca^2+^ peaks per astrocyte in all slices (white bars, *n* = 35 slices, two-tailed paired t test) and in slices from only female or male mice (*n* = 17 and *n* = 18, respectively; two-way RM ANOVA with Bonferroni’s multiple comparison test; sex, ns; treatment, *p* < 0.001; no interaction). (**e**) Representative example of astropile Ca^2+^ events extracted by AQuA (randomly colored), after each consecutive step of filtering, and merge of all filtered events, at basal conditions and during the 1st NE application. Thresholds for the filtering applied by area, amplitude, and duration are indicated on the top. Scale bar, 20 µm. (**f**) Same as in (**c**) for the frequency of Ca^2+^ events in the astropile (white bars, *n* = 35 slices, two-tailed paired t test) and in slices from only female or male mice (*n* = 17 and *n* = 18, respectively; two-way RM ANOVA with Bonferroni’s multiple comparison test; sex, ns; treatment, *p* < 0.001; no interaction). *p* < 0.001 (***), not significant (ns).

**Figure 3 cells-14-00024-f003:**
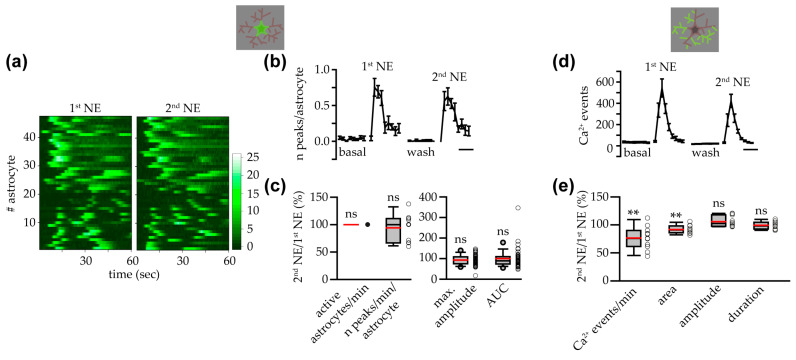
During a second challenge with NE, the astrocyte response was highly preserved. (**a**) Heatmaps of the ∆F/F_0_ in the soma of all astrocytes analyzed during the 1st and 2nd challenging with NE. (**b**) Time course of the mean number of somatic Ca^2+^ peaks per astrocyte in the different experimental conditions (*n* = 11 slices with 47 astrocytes from 7 mice). Scale bar, 1 min. (**c**) Box and whisker plots of the percentages of active astrocytes and frequency of somatic peaks per astrocyte in each slice (*n* = 11 slices with 47 astrocytes from 7 mice; two-tailed one-sample Wilcoxon signed rank test and two-tailed one-sample t test, respectively) and maximum amplitude and AUC in each responsive astrocyte (*n* = 47 astrocytes from 7 mice; two-tailed one-sample t test and two-tailed one-sample Wilcoxon signed rank test, respectively). To avoid excessive shrinkage of the data, in the following figures reporting AUC without antagonists, we do not represent the individual value of 345%. (**d**) Time course of the mean number of astropile Ca^2+^ events in the different conditions. Scale bar, 1 min. (**e**) Box and whisker plot of the percentages of the frequency, mean area, mean amplitude, and mean duration of astropile Ca^2+^ events (*n* = 11 slices; two-tailed one-sample *t* test: frequency, mean area and mean duration; one-sample Wilcoxon signed rank test: mean amplitude). *p* < 0.01 (**), not significant (ns).

**Figure 4 cells-14-00024-f004:**
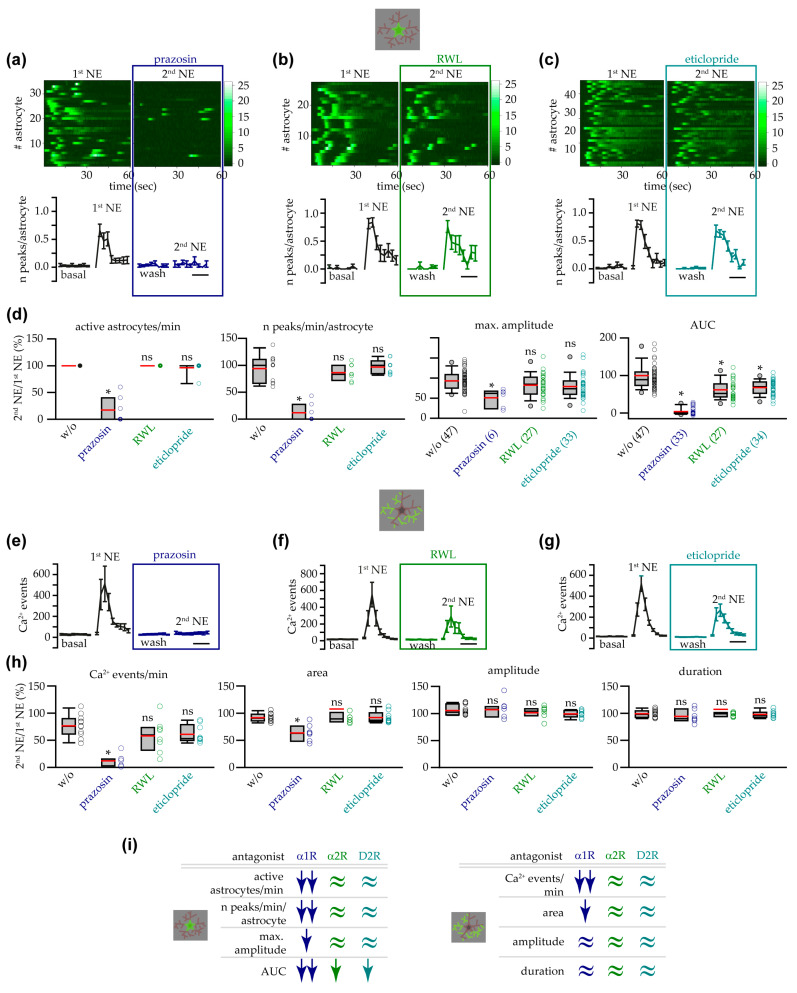
α1ARs mediated VTA astrocyte Ca^2+^ responses to NE. (**a**) Upper panel, heatmaps showing the ∆F/F_0_ in the soma of all astrocytes analyzed in the absence and in the presence of the α1-α2AR antagonist prazosin (10 μM). Lower panel, time course of the mean number of somatic Ca^2+^ peaks per astrocyte under different conditions (*n* = 7 slices with 35 astrocytes from 5 mice). Scale bar, 1 min. (**b**) Same as in (**a**) for the α2AR antagonist RWL (0.5 μM, *n* = 8 slices with 27 astrocytes from 5 mice). (**c**) Same as in (**a**) for the D2-type dopamine receptor antagonist eticlopride (1 μM, *n* = 9 slices with 35 astrocytes from 6 mice). (**d**) Box and whisker plots showing the percentages of active astrocytes/min and frequency of somatic peaks per astrocyte (slices: *w*/*o* antagonist, *n* = 11; prazosin, *n* = 7; RWL, *n* = 8; etilopride, *n* = 9; one-way ANOVA on ranks with post hoc test with Dunn’s method for comparison with the group *w*/*o* antagonist) and maximum amplitude and AUC (numbers reported in the treatment label indicate the number of astrocytes analyzed: max. amplitude, astrocytes responsive to both the 1st and 2nd NE; AUC, astrocytes responsive at least to the 1st NE; one-way ANOVA with Bonferroni test for comparison with the group *w*/*o* antagonist: maximum amplitude; one-way ANOVA on ranks with post hoc test with Dunn’s method for comparison with the group *w*/*o* antagonist: AUC). (**e**) Time course of the mean number of astropile Ca^2+^ events during NE application in the absence or presence of prazosin (*n* = 7 slices). Scale bar, 1 min. (**f**) Same as in (**e**) for the α2AR antagonist RWL (*n* = 8 slices). (**g**) Same as in (**e**) for the D2-type dopamine receptor antagonist eticlopride (*n* = 9 slices). (**h**) Box and whisker plots showing the percentages of the frequency, mean area, mean amplitude, and mean duration of the astropile Ca^2+^ events, in the presence of the different antagonists (*w*/*o* antagonist, *n* = 11; prazosin, *n* = 7; RWL, *n* = 8; eticlopride, *n* = 9; one-way ANOVA with Bonferroni test for comparison with the group *w*/*o* antagonist: frequency, mean amplitude; one-way ANOVA on ranks with post hoc test with Dunn’s method for comparison with the group *w*/*o* antagonist: mean area, mean duration). (**i**) Summary of the pharmacological experiments showing that VTA astrocyte Ca^2+^ responses to NE, at soma (**left**) and processes (**right**), were mediated mainly by α1ARs. *p* < 0.05 (*), not significant (ns).

**Figure 5 cells-14-00024-f005:**
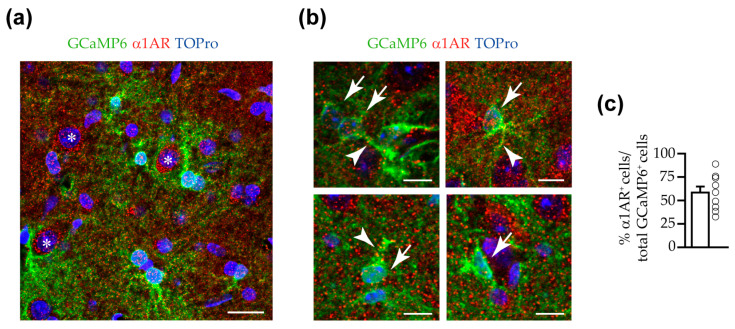
Expression of α1ARs in VTA astrocytes. (**a**) Confocal merged image of the VTA from a mouse injected with AAV5.GfaABC1D.cytoGCaMP6f.SV40, showing the green fluorescence of GCaMP6f (α-GFP), nuclear TOPro3 (blue), and the specific red staining for the α1AR. For the sake of illustration, the image shown is the z-projection of a 2.8 μm z-stack. Note the punctate staining of α1ARs in large cells not expressing GCaMP6f (white asterisk, presumed dopaminergic neurons) and, throughout the entire field of view, in regions outside cellular somata. Scale bar, 20 µm. (**b**) Merged images of single confocal planes, showing the staining for the α1AR in the soma (white arrows) and thick proximal processes (white arrowheads) of GCaMP6f-expressing astrocytes. Scale bar, 10 µm. (**c**) Bar chart showing the percentage of GCaMP6f^+^ cells that are positive for the α1AR in the soma and/or thick proximal processes. *n*  =  439 GCaMP6f^+^ cells from three mice, 5 slices, 28 fields. *p* < 0.05 (*).

**Figure 6 cells-14-00024-f006:**
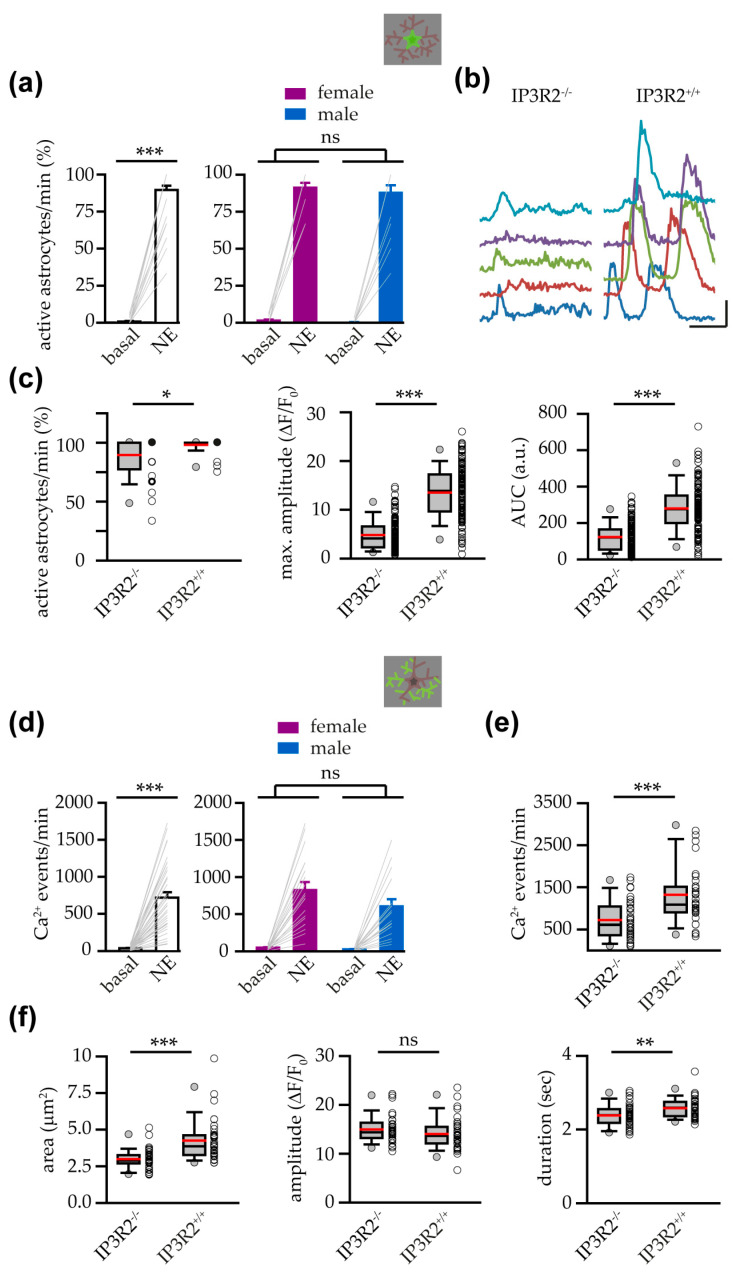
Astrocytes from IP3R2^−/−^ mice showed a reduced response to NE. (**a**) Bar charts showing the percentage of active astrocytes/min, at basal conditions and during the 1st NE application, in all the slices investigated from IP3R2^−/−^ mice (white bars, *n* = 37 slices with 149 astrocytes from 22 mice; Wilcoxon signed rank test) and in slices from only female (*n* = 19 slices with 77 astrocytes from 12 mice) or male IP3R2^−/−^ mice (*n* = 18 slices with 72 astrocytes from 10 mice) (Aligned Rank Transformation RM two-way ANOVA with Bonferroni multiple comparison test; sex, ns; treatment, *p* < 0.001; no interaction). (**b**) Representative examples of somatic Ca^2+^ dynamics (randomly colored), observed after stimulation with NE, in VTA astrocytes from IP3R2^+/+^ and IP3R2^−/−^ mice. Scale bars, 5 ∆F/F_0_; 20 s. (**c**) Box and whisker plots showing the NE response of VTA astrocytes from IP3R2^+/+^ and IP3R2^−/−^ mice, in terms of the percentage of active astrocytes/min (slices: IP3R2^+/+^ mice, *n* = 35 from 20 mice; IP3R2^−/−^, *n* = 37 from 22 mice; Mann–Whitney rank sum test), maximum amplitude (responsive astrocytes: IP3R2^+/+^ mice, *n* = 141 from 20 mice; IP3R2^−/−^, *n* = 132 from 22 mice; Mann–Whitney rank sum test), and AUC (responsive astrocytes: IP3R2^+/+^ mice, *n* = 141 from 20 mice; IP3R2^−/−^, *n* = 132 from 22 mice; Mann–Whitney rank sum test). (**d**) Same as in (**a**) for the frequency of Ca^2+^ events in the astropile of IP3R2^−/−^ mice (white bars, *n* = 37 slices, two-tailed paired t test) and in slices from only female or male IP3R2^−/−^ mice (*n* = 19 and *n* = 18 slices, respectively; two-way RM ANOVA with Bonferroni multiple comparison test; sex, ns; treatment, *p* < 0.001; no interaction). (**e**) Box and whisker plot showing the frequency of astropile Ca^2+^ events in slices from IP3R2^+/+^ and IP3R2^−/−^ mice, challenged with NE (slices: IP3R2^+/+^ mice, *n* = 35 from 20 mice; IP3R2^−/−^, *n* = 37 from 22 mice; Mann–Whitney rank sum test). (**f**) Box and whisker plots showing the mean area, mean amplitude, and mean duration of astropile Ca^2+^ events in slices from IP3R2^+/+^ and IP3R2^−/−^ mice, challenged with NE (slices: IP3R2^+/+^ mice, *n* = 35 from 20 mice; IP3R2^−/−^, *n* = 37 from 22 mice; Mann–Whitney rank sum test: area, amplitude; two-tailed *t* test, duration). *p* < 0.05 (*), *p* < 0.01 (**), *p* < 0.001 (***), not significant (ns).

**Figure 7 cells-14-00024-f007:**
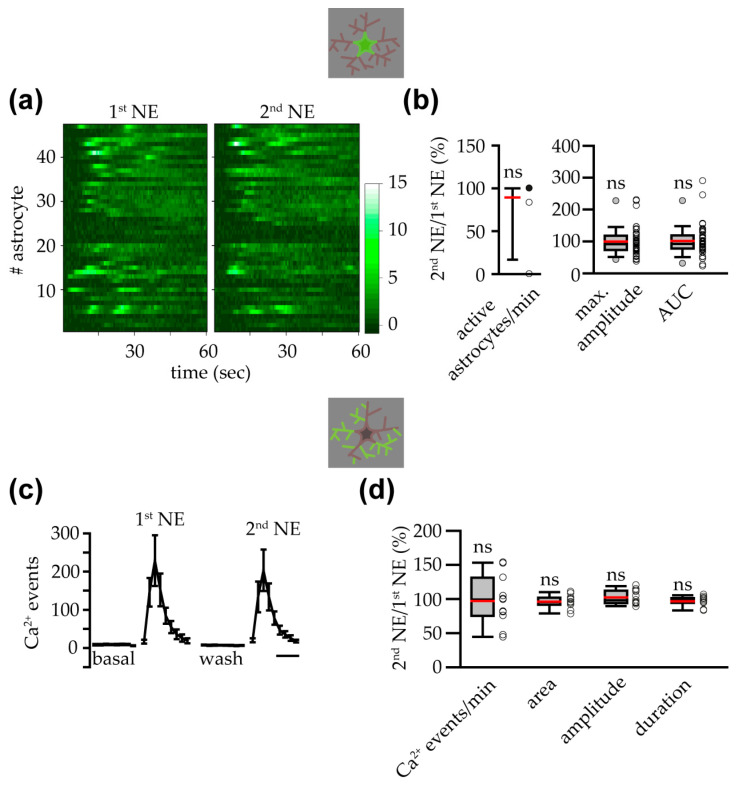
During a second challenge with NE, the response was highly preserved in astrocytes from IP3R2^−/−^ mice. (**a**) Heatmaps showing the ∆F/F_0_ in the soma of all astrocytes analyzed during the 1st and 2nd challenging with NE. (**b**) Box and whisker plots of the percentages of the active astrocytes/min (*n* = 11 slices with 47 astrocytes from 7 mice; two-tailed one-sample Wilcoxon signed rank test), maximum amplitude (*n* = 40 responsive astrocytes to both 1st and 2nd NE; two-tailed one-sample Wilcoxon signed rank test), and AUC (*n* = 43 responsive astrocytes at least to the 1st NE challenging; two-tailed one-sample Wilcoxon signed rank test). (**c**) Time course of the mean number of astropile Ca^2+^ events in the different experimental conditions. Scale bar, 1 min. (**d**) Box and whisker plot of the percentages of the frequency, mean area, mean amplitude, and mean duration of the astropile Ca^2+^ events (*n* = 11 slices; two-tailed one-sample *t* test). not significant (ns).

**Figure 8 cells-14-00024-f008:**
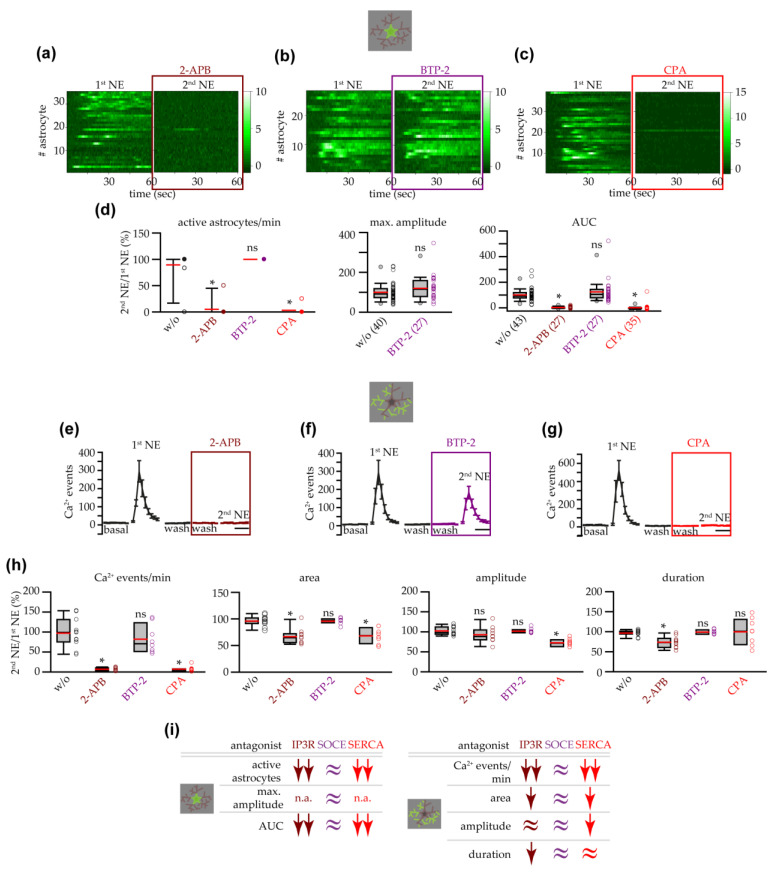
NE-triggered astrocyte response in IP3R2^−/−^ mice depended on ER intracellular Ca^2+^ stores and IP3Rs. (**a**) Heatmaps showing the ∆F/F_0_ in the soma of all astrocytes analyzed, in the absence or in the presence of the IP3R2 receptor antagonist 2-APB (100 μM). (**b**) Same as in (**a**) for the SOCE antagonist BTP-2 (25 μM). (**c**) Same as in (**a**) for the SERCA pump antagonist CPA (50 μM). (**d**) Box and whisker plots showing the percentages of active astrocytes/min (slices: *w*/*o* antagonist, *n* = 11 slices with 47 astrocytes from 7 mice; 2-APB, *n* = 10 slices with 35 astrocytes from 7 mice; BTP-2, *n* = 8 slices with 28 astrocytes from 7 mice; CPA, *n* = 8 slices with 39 astrocytes from 4 mice; one-way ANOVA on ranks with post hoc test with Dunn’s method for comparison with the group *w*/*o* antagonist), and maximum amplitude and AUC (numbers reported in the treatment label indicate the number of astrocytes analyzed: max. amplitude, astrocytes responsive to both the 1st and 2nd NE (note that 2-APB and CPA groups were not included because only 1 astrocyte displayed activity during both the 1st and 2nd NE); AUC, astrocytes responsive at least to the 1st NE; Mann–Whitney rank sum test: maximum amplitude; one-way ANOVA on ranks with post hoc test with Dunn’s method for comparison with the group *w*/*o* antagonist: AUC). (**e**) Time course of the mean number of astropile Ca^2+^ events extracted during NE application in the absence or presence of 2-APB (*n* = 10 slices). Scale bar, 1 min. (**f**) Same as in (**e**) for the BTP-2 antagonist (*n* = 8 slices). (**g**) Same as in (**e**) for the CPA antagonist (*n* = 8). (**h**) Box and whisker plots showing the percentages of the frequency, mean area, mean amplitude, and mean duration of astropile Ca^2+^ events, in the presence of the different antagonists (*w*/*o* antagonist, *n* = 11; 2-APB, *n* = 10; BTP-2, *n* = 8; CPA, *n* = 8; one-way ANOVA on ranks with post hoc test with Dunn’s method for comparison with the group *w*/*o* antagonist: frequency, amplitude, duration; one-way ANOVA with Bonferroni test for comparison with the group *w*/*o* antagonist: area). (**i**) Summary of the pharmacological experiments showing that VTA astrocyte Ca^2+^ responses, at soma (left) and processes (right), to NE in IP3R2^−/−^ mice were mediated mainly by ER intracellular Ca^2+^ stores and IP3Rs. n.a., not applicable. *p* < 0.05 (*), not significant (ns).

## Data Availability

Data generated and/or analyzed during the current study are available from the corresponding author upon reasonable request.

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
