# Peer review of "Characterization of the Astrocyte Calcium Response to Norepinephrine in the Ventral Tegmental Area"

_cells, 2024, doi:10.3390/cells14010024_

Round 1
Reviewer 1 Report
Comments and Suggestions for Authors
In this study, the authors demonstrate that astrocytes in the Ventral Tegmental Area (VTA) are new targets for noradrenergic (NE) signaling. The results illustrate how NE activates the glial cells, specifically astrocytes, within the VTA circuit. This highlights the need for new interpretations of the cellular and molecular mechanisms underlying the effects of NE in this neuronal circuit. The authors present a complex methodology that is well explained. The results are well-argued and thoroughly discussed. The figures are clear, well-described, and effectively recap the obtained results. The references are consistent and appropriate. The originality and scientific soundness of this study are high. This study can be expected to stimulate active discussion among a wide range of Cells readers.
Author Response
Open Review1
In this study, the authors demonstrate that astrocytes in the Ventral Tegmental Area (VTA) are new targets for noradrenergic (NE) signaling. The results illustrate how NE activates the glial cells, specifically astrocytes, within the VTA circuit. This highlights the need for new interpretations of the cellular and molecular mechanisms underlying the effects of NE in this neuronal circuit. The authors present a complex methodology that is well explained. The results are well-argued and thoroughly discussed. The figures are clear, well-described, and effectively recap the obtained results. The references are consistent and appropriate. The originality and scientific soundness of this study are high. This study can be expected to stimulate active discussion among a wide range of Cells readers.
We thank the reviewer for the positive comments on our manuscript, particularly for stating that it has the potential to stimulate active discussion.
Reviewer 2 Report
Comments and Suggestions for Authors
In this paper, Speggiorin and colleagues report on their study of calcium response in astrocytes exposed to norepinephrine (NE) on mice brain slices with a focus on the VTA. This was elegantly conducted using a genetically encoded calcium indicator expressed in vivo after stereotaxic injection of a dedicated AAV construct. Expression was restricted to astrocytes thanks to the specific promoter GfaABC1D. Fluorescence signals were monitored and recorded using either a confocal laser microscope or a two-photon laser scanning microscope. The paper provides a comprehensive analysis of the signal triggered by exposing the tissue to NE in the cell somata or in astrocytes processes. Using a pharmacological approach (antagonists of adrenergic receptor subtypes or modulators of cell calcium homeostasis) or using transgenic animals lacking the IP3R2, the authors have generated convincing data suggesting that the response in astrocytes was essentially driven by alpha1 adrenoreceptors and a release of endoplasmic reticulum stored calcium.
The data are well presented and largely support the conclusion drawn by the authors. The following comments however deserve to be taken into account to consolidate the quality of the study and particularly the presentation.
- The paper is long and suffers from extensive repetition of the experimental paradigm in the M&M section and in the description of the results (and even a third time in the legends of the figures). Also, there is a large replication of the results detailed both in the main text and in the legends of the figures. Hence all data from histograms in the figures are systematically repeated in the text by detailing all values (means and sem). This makes the text really unclear for readers.
- The rationale for systematically repeating the experiment -and to provide all results details on the data- on either male or female is far from obvious. Is there any reason to hypothesise that there might be some difference. Indeed, this is not pointed in the introduction and poorly discussed in the discussion section (even though the authors consider this finding as remarkable). All this tends to dilute the key messages of the paper.
- Similarly, the reason to specifically and deeply address the question of the role of IP3R2 in the response to NE in VTA astrocytes remain obscure. As presented, the paper turns rather descriptive or even technical but does not explain the scientific advances regarding the role of alpha1 adrenoreceptors in VTA astrocytes or the importance of involving a specific IP3 receptor.
- In figure 1, panel b raises questions as the fluorescence images poorly correlate with the histograms in c. Indeed, the GcAMP6 expression seems to be highly variable from cells to cells with a few cells strongly positive. Staining with S100B is largely diffused and thus it seems difficult to validate that the majority of S100B labelled cells express the calcium probe. Maybe better-quality images could be more convincing. As the probe is not a ratiometric indicator, can we ensure that the recorded response is not influenced by the expression level of the probe?
- Do the authors check whether S100B stains the same cells as SR101?
- The authors claim that the response to NE is dose dependent (one would suggest to use the term concentration-dependent instead). But only two concentrations were tested, which makes it weak for such strong interpretation of dose-dependency . Also, with respect to the pharmacological approach, the authors should explain the rationale for the competitive antagonist concentrations tested (which are only communicated discretely in the legend of figure 4, line585). How can we assure selectivity and sufficient antagonism with these concentrations, in particular when knowing that the affinity of NE for the different receptor candidates is not the same (see discussion, line 854).
- While all astrocytes tend to show positive response, only half of them appear to express the targeted alpha1 receptor. The authors suggest a spreading of the calcium signal from processes to cell soma. But this does not explain how all cells respond positively. Is there any chance for a cx43-dependent signal spreading between neighbour cells?
- Other GPCR coupled to Gq are expressed in astrocytes (GroupI mGlu for instance) and it would be worth to compare the data obtained with NE and with another transmitter.
Reviewer 3 Report
Comments and Suggestions for Authors
Speggiorin Cells 2024
The study investigated Ca2+ dynamics in VTA astrocytes and their responses to noradrenaline. The authors report variable Ca2+ events after application of NA 1-10 mkM, which they could block with an alpha1 adrenergic antagonist. They also found that in IP3R KO mice these responses were partially preserved and that Ca2+ originated from an internal store. Finally they report no sex differences in these responses.
As such the study is carefully done, data is analysed and described in great detail and text is mainly well written with some elements which could be trimmed out because they a somewhat repetitive.
My main criticism is that there is hardly anything new and exciting about Ca2+ responses in astrocytes, caused by application of NA. The authors cite several studies where it was shown including data from in vivo 2P microscopy. So, except that this is now shown in VTA, what is new here? There are lots of different nuclei in the brain, do we need a separate study on each one of them? Was there any reason to anticipate something unusual here? The same goes for the comparison of males and females. No difference was found but why was that even questioned?
Finally, it is a major mistake to believe that monitoring Ca2+ gives you the way to assess the effects of almost anything on astrocytes. The filed has been inundated by hundreds of papers where all sorts of Ca2+ events in astrocytes were described. But what is the take home message? If Ca2+ in an astrocyte goes up/down, what will that do? Ca2+ dependent exocytosis from astrocytes in vivo is a highly debatable issue, but even if it does exist and is significant, the real question is what does it do to the function of the network. However, and this is important, the cAMP is definitely number one pathway to look at in astrocytes, especially in connection to NA, because they express more beta AR than alpha and because increases in cAMP in astrocytes have obvious biological implications. Therefore the authors, having spent a lot of time on watching intracellular Ca2+ (as lots of other groups do), almost certainly missed other effects of NA via cAMP. Obviously, the study is what it is, and the authors want to publish it, but they have to at least acknowledge that they only looked at one of several intracellular signalling pathways in astrocytes, and not necessarily the most relevant to the action of NA.
BTW, they cite a study where residual Ca2+ responses in astrocytes from IP3R KO mice were linked to Ca2+ release from mitochondria Okubo et al). That paper has data, basically very similar to the current manuscript, so what is really new here?
To summarise, the authors should try to better explain what is the novelty of their study and why they think it is important to show that NA can trigger Ca2+ responses in astrocytes there. Plus, they need to acknowledge that they only looked at Ca2+ which is definitely only a fraction of the whole picture. Some recent papers show quite clearly how important cAMP in astrocytes is.
Some specific comments
41 are associated to diverse intracellular – associated WITH
65 In accordance with this preferred signaling through – preferential
85-86 However, whether astrocyte from the VTA - ASTROCYTES
147 NE was then washed out and, during - for how long?
194 In the experiments with CPA, the application of the antagonist initiated - WAS initiated
210 – 215 and other instances. Do the authors not see any evidence of desensitisation for repeated administration of NA? How long did they need to wait to get the 2nd response exactly like the first one, was it even possible? Can the authors show 2 or 3 sequential responses to 10 uM NA?
293-312 – This text is largely redundant, all of that is in the Methods.
Fig 1B -suggested to put arrows at the co-localised cells.
379-380 – This is not accurate, Pankratov used nanomols.
388 Astrocyte response to NE at soma and processes is similar in female and male mice – but what was the reason to look for differences?
410 Figure 2c shows that, in the presence of TTX, almost all astrocytes –
Use of TTX is not a guarantee that neurones are not involved. GPCRs such as NA receptors can activate release of transmitters without action potentials, especially from somato-dendritic compartments .
Figure 5. There seems to be a lot of nuclear staining, this is alarming. How good is this AB?
804 – 805 – “with 100 M ATP and the Ca2+ responses elicited in astrocytes, even if present, were dras-804 tically lower compared to the response evoked by NE in the same astrocytes (data not 805 shown).” – I would like to see these results (supplement?), This extremely odd, because ATP is usually the strongest stimulant for astrocytes in Ca2+ essays.
807-808 – Why was it “remarkable”? Why was a difference expected?
Round 2
Reviewer 2 Report
Comments and Suggestions for Authors
The authors have addressed all comment raised from the original submission. The corrections are appreciated.